# Formation of Acrylamide and Other Heat-Induced Compounds during Panela Production

**DOI:** 10.3390/foods9040531

**Published:** 2020-04-22

**Authors:** Marta Mesias, Cristina Delgado-Andrade, Faver Gómez-Narváez, José Contreras-Calderón, Francisco J. Morales

**Affiliations:** 1Instituto de Ciencia y Tecnología de Alimentos y Nutrición (ICTAN-CSIC), C/José Antonio Novais, 10, E-28040 Madrid, Spain; mmesias@ictan.csic.es (M.M.); fjmorales@ictan.csic.es (F.J.M.); 2Bioali Research Group, Food Department, Faculty of Pharmaceutical and Food Sciences, University of Antioquia, Calle 67 No. 53–108, Ciudad Universitaria, Medellín 050010, Colombia; faver.gomez@udea.edu.co (F.G.-N.); jose.contrerasc@udea.edu.co (J.C.-C.)

**Keywords:** panela, processing, acrylamide, furanic compounds, antioxidants, non-enzymatic browning

## Abstract

Non-centrifugal cane sugar (panela) is an unrefined sugar obtained through intense dehydration of sugarcane juice. Browning, antioxidant capacity (measured by ABTS (2,2’-azino-bis (3-ethylbenzothiazoline-6-sulfonic acid) assay and total phenolic content) and the formation of acrylamide and other heat-induced compounds such as hydroxymethylfurfural (HMF) and furfural, were evaluated at different stages during the production of block panela. Values ranged between below the limit of quantitation (LOQ)–890 µg/kg, < LOQ–2.37 mg/kg, < LOQ–4.5 mg/kg, 0.51–3.6 Abs 420 nm/g, 0.89–4.18 mg gallic acid equivalents (GAE)/g and 5.08–29.70 µmol TE/g, for acrylamide, HMF, furfural, browning, total phenolic content and ABTS (all data in fresh weight), respectively. Acrylamide significantly increased as soluble solid content increased throughout the process. The critical stages for the formation of acrylamide, HMF and furfural were the concentration of the clarified juice in the concentration stage to get the panela honey and the final stage. Similar trends were observed for the other parameters. This research concludes that acrylamide, HMF and furfural form at a high rate during panela processing at the stage of juice concentration by intense evaporation. Therefore, the juice concentration stage is revealed as the critical step in the process to settle mitigation strategies.

## 1. Introduction

Production of panela, also known as unrefined non-centrifugal sugar or non-centrifugal cane sugar, is one of the most traditional agro-industries in tropical countries. Panela is obtained by grinding the sugar cane, clarifying, evaporating the juice and concentrating it until honey is obtained (more than 90° Brix). This is then beaten, molded and cooled to achieve solidification [1]. Panela production exhibits yields between 6.4% and 14.9% [2].

Panela is produced by small farmers around the world. India is by far the world’s largest producer of panela, accounting for 56.15% of global production in 2011. Colombia, the world’s second largest producer, contributed 14.1% of total panela globally in 2011. While India and Colombia dominate global production, other countries from Asia, Africa or Latin America are also producers of this foodstuff [3]. The use of panela in industrially produced foods is as a sweetener to replace refined sugars, but also as an ingredient for the manufacture of foodstuffs such as puddings, baked goods, marmalades, protein/energy/weight control bars, desserts, confectionary and chocolate products [4].

Unlike white sugar, panela contains minerals, vitamins, phenolic compounds, amino acids and proteins. The presence of bioactive, health-promoting compounds increases the beneficial effects of panela on human health, including effects that are anticarcinogenic, antitoxic, cytoprotective, anti-inflammatory and antiatherogenic [5].

Due to high sugar (≈14 g/100 g) and nitrogen compound (≈0.40 g/100 g) levels in sugarcane [6], the Maillard reaction and caramelization are the main chemical reactions to take place during panela production. The Maillard reaction is promoted at >50 °C and pH 4–7 and occurs in low moisture conditions. The initial step involves the formation of a Schiff base from the reaction of the amino group on an amino acid with the carbonyl group of a reducing sugar, which can rearrange to form an Amadori compound whose degradation yields intermediate compounds such as acrylamide and furanic compounds. Caramelization is favored at temperatures of >120 °C and pH 3–9 and involves another nonenzymatic browning step through the degradation of reducing sugars without the condensation step [7]. Acrylamide formation in panela is possible due to the presence of free amino acids and reducing sugars, intense thermal treatment, and the low moisture found in the final product [1,6]. Acrylamide has been described to present neurotoxic, genotoxic, carcinogenic and reproductive toxic effects [8,9]. On the other hand, furanic compounds such as hydroxymethylfurfural (HMF) and furfural have been extensively applied as heat-induced chemical indexes for monitoring the thermal damage of food. These furan derivatives, which can be generated during panela production through both caramelization and Maillard reactions [7,10], have been confirmed to confer genotoxic, mutagenic, carcinogenic, DNA-damaging, organotoxic and enzyme inhibitory effects. It has been reported that HMF is an indirect mutagen because it is converted to an active metabolite, the sulfuric acid ester 5-sulfo-oxymethylfurfural (SMF), with mutagenicity [11], whereas furfural has shown toxicological effects leading to hepatotoxicity [12].

Scientific research is scarce in relation to the occurrence and pathways involved in the formation of heat-induced process contaminants, particularly acrylamide and furfurals, during panela production. In addition, data have mostly been derived from methodologies that apply analytical techniques which lack sensitivity and fail to refer to normalized procedures [1]. Due to the toxicological effects on human health related to these compounds, evaluation of their formation in foods is necessary to look for mitigation strategies aimed to reduce the exposure to the contaminants. Thus, the aim of this work was to study the impact of different stages during the panela production chain on the formation of acrylamide and other heat-induced process contaminants by applying robust and validated analytical methods. The results will provide greater insight into the identification of critical points in the process. It is the first time that such assays have been carried out together and, therefore, the outcomes of this investigation could be key for establishing mitigation strategies which can be used by panela producers and food safety bodies.

## 2. Material and Methods

### 2.1. Chemicals and Reagents

Potassium hexacyanoferrate (II) trihydrate (98%, Carrez-I) and zinc acetate dehydrate (>99%, Carrez-II) were obtained from Sigma (St. Louis, MO, USA). ^13^C_3_-labelled acrylamide (99% isotopic purity) was obtained from Cambridge Isotope Laboratories (Andover, MA, USA. Formic acid (98%), D(+) glucose, D(-) fructose, D(-) sorbitol, ethanol, methanol (99.5%) and hexane were obtained from Panreac (Barcelona, Spain). Deionized water was obtained from a Milli-Q Integral 5 water purification system (Millipore, Billerica, MA, USA). All other chemicals, solvents and reagents were of analytical grade.

### 2.2. Samples

Samples were supplied by a large panela producer located in the province of Antioquia (Colombia). Samples were collected at four different stages, which represent critical points during the manufacture of block panela (Figure 1). Samples were as follows: raw cane juice obtained after the cane is ground (Sample 1); clarified juice obtained by heating (<100 °C) sample 1 (Sample 2); concentrated juice produced at 60–65° Brix following evaporation (>110 °C) (Sample 3); block panela obtained following concentration (>120 °C) (Sample 4). Evaporation of the juice is performed in batches of 1000 L, whilst being heated at 110–120 °C for approximately 40 min. Panela honey concentration is performed at 120 °C for approximately 20 min in batches of 40 L. Samples 1, 2 and 3—which correspond to the juices—were lyophilized and stored at −20 °C whilst awaiting analysis. Block panela (sample 4) was stored at room temperature.

### 2.3. Basic Analyses

Moisture was estimated gravimetrically to constant weight following well-established procedures for panela juices [13] and block panela [14]. The pH of panela was determined by mixing the sample (1 g) with 100 mL of water and vortexing for 3 min. The mixture was kept at room temperature for 1 h and centrifuged to separate impurities. The pH was measured using a CG-837 pH meter (Schott, Mainz, Germany). The pH of juices was directly measured in fresh samples immediately following collection. Soluble solids content (°Brix; in original juices immediately after their collection) were measured with a digital refractometer OptiDuo (Bellingham + Stanley, Kent, UK). Soluble solid content in panela was calculated from the moisture content.

### 2.4. Determination of Reducing Sugars

The content of reducing sugars (glucose + fructose) was determined in lyophilized juices and in the panela sample via high performance liquid chromatography using a refractive index detector (HPLC-RID). The procedure was based on a slightly modified version of the method described by Ayvaz [15]. Three hundred mg of sample was weighed and mixed with 9 mL of 80% (*v*/*v*) ethanol and 1 mL of sorbitol (10 mg/mL), as an internal standard. Following vortex agitation, the mixture was incubated at 50 °C and 900 rpm for 1 h, and centrifuged at 4 °C and 5000 rpm for 20 min. The supernatant was transferred into a new tube and ethanol was evaporated using TurboVap equipment (Biotage, Uppsala, Sweden). The aqueous extract was purified via solid-phase extraction using an SCX cartridge (Supelco, Sigma Aldrich, St. Louis, MO, USA) and filtered (0.22 µm pore-size membrane) prior to HPLC analysis. Twenty µL of extract was injected into the HPLC System LC-20 AD, using a RID-10A (Shimadzu, Scientific Instruments, INC, Columbia, MD, USA). Analytical separation was achieved with a Rezex RCM-Monosaccharide Ca^2+^ column (300 × 7.8 mm, 8 µm; Phenomenex, Torrance, CA, USA) at 80 °C in isocratic elution, with a mobile phase of deionized water and a flow rate of 0.6 mL/min. Sugars were quantified using standard solutions spiked with sorbitol. Results were expressed as g/100 g of fresh weight (FW) and dry matter (DM). The analysis was performed in duplicate.

### 2.5. Determination of Asparagine

Asparagine was determined in the lyophilized juices and panela sample via gas chromatography-flame ionization detection (GC-FID), according to Farkas and Toulouee [16] but with some minor modifications as described by Mesias et al. [17]. A GC-FID (Agilent GC 7820A FID) equipped with an automatic injector was used for quantitation. An amino acid dedicated column (Zebron ZBAAA capillary; 10 × 0.25 mm) was used to separate amino acids. Starting oven temperature was set at 110 °C and increased 32 °C per minute until 320 °C was reached. An aliquot of the derivatized sample (1 µL) was injected in split mode (15:1) at 250 °C. The FID detector was set to 320 °C and the carrier helium gas flow rate was maintained at 1.5 mL/min whilst in process. External calibration was carried out using asparagine standard and results were corrected according to norvaline recovery, this being used as an internal standard. Free asparagine content was expressed as mg/100 g of FW and DM. Analysis was performed in duplicate.

### 2.6. LC-ESI-MS-MS Determination of Acrylamide

Sample extraction followed a slightly modified version of the method described by Mesias and Morales [18]. Lyophilized juice and panela samples (0.5 g) were weighed and mixed with 9.4 mL of water in polypropylene centrifugal tubes. The mixture was spiked with 100 µL of a 5 µg/mL [^13^C_3_]-acrylamide methanolic solution, which served as an internal standard, and later homogenized (Ultra Turrax, IKA, Mod-T10 basic, Bohn, Germany) for 10 min. Afterwards, samples were treated with 250 µL of Carrez I (15 g potassium ferrocyanide/100 mL water) and Carrez II (30 g zinc acetate/100 mL water) solutions, and centrifuged (9000 g for 10 min) at 4 °C. Samples were clarified using Oasis-HLB cartridges (Supelco, Saint Louis, MO, USA) and extracts were analyzed according to Mesías and Morales [18]. Acrylamide recovery occurred between 90% and 106%. Relative standard deviations (RSD) for precision, repeatability and reproducibility of the analyses were calculated as 2.8%, 1.2% and 2.5%, respectively. The procedure fulfilled method performance requirements established by the EU acrylamide Regulation 2017/2158. The limit of quantitation was set at 20 µg/kg. Acrylamide results were expressed as µg/kg of FW and DM, and comparisons were made between different stages. Samples were analyzed in duplicate.

### 2.7. Determination of HMF and Furfural

HMF and furfural content was determined in lyophilized juices and panela samples using High-Performance Liquid Chromatography with Diode-Array Detection (HPLC-DAD), as described by Mesías et al. [19]. The limits of quantification were set at 0.6 and 0.3 mg/kg for HMF and furfural, respectively. Results were expressed as mg/kg of FW and DM. Samples were analyzed in duplicate.

### 2.8. Determination of Browning

Supernatant fractions (200 µL) obtained during preparation of HMF and furfural samples were placed in 96-well plates. Browning at 420 nm was measured at room temperature using a BioTekSynergyTM HT-multimode microplate spectrophotometer (BioTek Instruments, Winooski, VT, USA). Samples were analyzed in duplicate and results were expressed as absorbance units (AU)/g of FW and DM.

### 2.9. Sample Extraction for Measurement of Antioxidant Activity (ABTS (2,2’-azino-bis(3-ethylbenzothiazoli-ne-6-sulfonic acid) Assay and Total Phenolic Content)

Sample extraction was performed following the procedure described by Pérez-Jiménez and Saura-Calixto [20]. Briefly, 0.1 g of lyophilized juice and panela sample was placed in a tube, and 6 mL of acidic methanol/water (50:50 *v*/*v*, pH 2) was added. The tube was thoroughly shaken at room temperature for 20 min and centrifuged at 2500 g for 10 min in order to recover the supernatant. Four milliliters of the same acidic methanol/water solution were added to the residue, with shaking and centrifugation steps then being repeated. The second methanolic extract was combined with the first one. When necessary, proper dilutions with distilled water were performed to measure in the ABTS assay and the total phenolic content. Extraction was performed in duplicate.

#### 2.9.1. Total Phenolic Content (TPC)

Total phenolic content was determined according to a slightly modified version of the Folin–Ciocalteu method described by Marfil et al. [21]. Briefly, 80 µL of sample, blank or gallic acid standard, 1520 µL of distilled water and 300 µL of 20% Na_2_CO_3_ (*w*/*v*) were mixed with 100 µL of commercial Folin–Ciocalteu’s reagent and incubated for 1h at room temperature. Absorbance was measured at 750 nm using a Biotek Synergy HT multi-mode microplate reader (BioTek^®^ Instruments Inc., USA). Results were expressed as mg gallic acid equivalent (GAE)/g of FW and DM. All measurements were performed in triplicate.

#### 2.9.2. ABTS Assay

The ABTS assay was developed as described by Rufián-Henares and Delgado-Andrade [22], with slight modifications. The ABTS+• was produced by reacting 7 mM ABTS stock solution with 2.45 mM potassium persulfate and allowing the mixture to stand in the dark at room temperature for 12–16 h before use. The ABTS+• working solution (stable for 2 days) was diluted with an ethanol: water (50:50) solution until an absorbance of 0.70 ± 0.02 at 730 nm was achieved. For analyses, 40 µL of sample, blank or Trolox standard and 200 µL of 5 mM pH 8.4 phosphate buffer were added to 60 µL of diluted ABTS+• solution. The absorbance reading was taken at 10 min using the previously described microplate reader. Aqueous Trolox solutions were used for calibration (15–125 µM). Results were expressed as µmol Trolox equivalents (TE)/g of FW and DM. All measurements were performed in triplicate.

### 2.10. Statistical Analysis

One-way ANOVA was used to investigate differences between final block panela and sugar cane juice, in the content of processing contaminants and other physicochemical variables at three different processing stages. Significant differences were established using the LSD test, with a confidence of 95%. All statistical analyses were performed using Statgraphics Centurion^®^ Version XVI (The Plains, VA, USA).

## 3. Results and Discussion

Samples collected at different stages during panela processing were analyzed for pH value, moisture and soluble solids content (Table 1). According to the good processing practices for high quality block panela [6], the pH of the mature sugar cane must increase from 5.0–5.3 to 5.2–5.4 after crushing (raw juice cane). Then, during the clarification stage, the pH of the juice will remain between 5.8 and 6.5 by the addition of pH regulators that will avoid sucrose inversion [6]. Similarly, the soluble solids in the raw juice must be ≥20° Brix, and between 18 and 20° Brix in the juice following clarification. After the heating and evaporation stage, the pH of the panela honey must be at least 5.8, with the soluble solids around 65–70° Brix. At the final stage, the panela normally exhibits pH values between 5.6 and 6.3, and a soluble solid content between 88° and 92° Brix. In the present study, results of pH and soluble solid content are in line with that described above for block panela. Sample 1 (raw juice) showed a pH value of 5.4, which increased to 6.0 following pH regulation. In the same way, soluble solid content increased from 21.2 in sample 1, to 92.4° Brix in the block panela. As expected, the moisture content decreased from 78.8 to 7.6% as panela elaboration progressed.

### 3.1. Acrylamide

The acrylamide content ranged from < limit of quantitation (LOQ) to 890 μg/kg sample (FW) (Table 2). Throughout the panela manufacturing progress, the formation of acrylamide increased from sample 1 to sample 4 (panela). This increase in acrylamide formation is in line with the use of higher temperatures and the relative concentration of solids in the samples. At the end of the concentration step (Figure 1), sample 3 (67.5° Brix) exhibited an acrylamide content of 298 µg/kg, with this climbing sharply to reach 890 µg/kg (Table 2) in the final product (92.4° Brix). In line with Vargas Lasso et al. [1], negligible amounts of acrylamide (<LOQ) were detected in raw cane juice and in the juice following clarification. Vargas Lasso et al. [1] reported higher values in concentrated juice (800 µg/kg) and final block panela (2200 µg/kg) than those observed in our study. The differences with these authors could be related to the different origins of sugar cane, amounts of acrylamide precursors and processing conditions. However, application of a non-specific and non-selective analytical technique for acrylamide should also be considered as this could lead to overestimations instead of detections based on tandem mass spectrometry. The fate of acrylamide in the samples corroborates the conclusion that temperature and moisture content are the key physical parameters to influence acrylamide formation during panela production, whilst maintaining levels of precursors in sugar cane.

Data available in the literature are scarce regarding acrylamide content in panela. Hoenicke and Gatermann [23] found values of around 140 μg/kg in raw sugar. These data are much lower than that detected in the present study. This may be attributed to reasons such as a more drastic thermal process, high content of precursors in raw juice and inadequate pH regulation, which favors sucrose hydrolysis followed by the Maillard reaction. The acrylamide level measured in the block panela sample was close to the highest values reported by Gómez-Narváez et al. [24] in eight panela blocks, with an average figure of 540 μg/kg.

Figure 2 shows the trends in acrylamide, asparagine and reducing sugars (results expressed as DM) at critical stages of the process. No significant changes (*p* > 0.05) were observed in sample 1 and sample 2 regarding acrylamide and its precursors. However, the acrylamide content increased, whilst the content of its precursors decreased during the concentration stage. It is unexpected that asparagine content increased in sample 3 (expressed in DM), however, standard deviations between measurements were much higher when compared to the other samples. Residual proteolysis activity following the breakage of vegetable cells is also plausible during crushing and clarification stages at moderate temperatures, prior to inactivation induced by temperatures higher than 100 °C and the resulting increase in asparagine. The concentration stage during panela production is characterized by intense heat treatment during which clarified juice is boiled at temperatures higher than 120 °C. Following this, a marked drop in asparagine was displayed, together with a high increase in the acrylamide concentration in block panela. With regard to the reducing sugars content, a trend towards a decrease is shown, with a steeper decline observed between sample 2 and sample 3. The stable levels seen in reducing sugars between sample 3 and sample 4 (panela) would suggest that acrylamide formation during this stage occurs through several reaction mechanisms. These mechanisms occur in low humidity conditions and involve decarboxylation of the Schiff base, leading to Maillard intermediates that directly or indirectly release acrylamide [25]. The reducing sugar content patterns in panela are similar to those reported by other authors. Jaffé [26] described levels of between 3.69 and 10.5 g/100 g, whereas Lee et al. [27] indicated ranges of 2.10–3.30 g/100 g for glucose and 1.76–2.56 g/100 g for fructose. These high reducing sugar levels suggest that asparagine could be the limiting factor of acrylamide formation in panela samples.

### 3.2. HMF, Furfural and Browning

The HMF and furfural contents ranged from <LOQ to 2.37 mg/kg of FW and from <LOQ to 4.50 mg/kg of FW (Table 2) in sample 1 and sample 4 (panela), respectively (Figure 3A,B). When expressed according to dry matter, values ranged from <LOQ to 2.57 mg/kg for HMF and from <LOQ to 4.87 mg/kg for furfural. Both HMF and furfural are chemical markers of progress in the heat process [22]. However, the values are low considering the high sugar content (sucrose) seen in sugarcane and the high temperatures applied during the process. These values are similar to those reported in milk proteins (not detected (ND)–7.42 mg HMF/kg) [28,29] and infant formulas (not detected–14.2 mg HMF/kg and not detected–0.62 mg Furfural/kg), which do not have such a high sugar content or undergo such drastic thermal processes. HMF and furfural content in panela was quite low in comparison with other food matrices such as coffee (23.3–4112 mg HMF/kg) [30], dried fruit (25–2900 mg HMF/kg) and balsamic vinegar (316.4–35251.3 mg HMF/kg) [31]. Gómez-Narváez et al. [24] found average HMF and furfural values in panela blocks of 5.9 and 3.0 mg/kg, respectively, which is similar to those found in the present study. Since furfural is generated from pentoses and not from hexoses, it may be hypothesized that furfural is formed from the interconversion of HMF. This occurs as a result of strong heating conditions [32], which is supported by the fact that furfural was only found in the final product (block panela).

Measurement of absorbance at 420 nm in the soluble fraction of panela is a parameter that is used to monitor the extent of browning reactions [33]. Several compounds account for the absorbance seen at 420 nm. These include natural compounds present in the juice such as phytochemicals, and those formed during processing such as products of caramelization, the Maillard reaction or the oxidation of phenolic compounds. Browning ranged from 0.51 to 3.60 AU/g of FW (Table 2) for sample 1 and sample 4, respectively. Browning in panela is similar to that reported by Gómez-Narváez et al. [24] in block panela (mean 2.2 units/g). In a similar way to HMF and furfural, significant differences were observed (*p* < 0.05) when different stages of the process in DM were considered (Figure 3). In the case of browning, the greatest increase occurred in panela (1.6 times the value detected in sample 1) (Figure 3C). This was an expected result given the progress of the thermal treatment applied during different stages of block panela production. In accordance with the findings relating to other parameters analyzed in the present study, no outcomes were seen in relation to the progress of browning during the processing of raw sugar cane juice.

### 3.3. Antioxidant Activity

TPC and ABTS ranged from 0.89 to 4.18 GAE/g of FW and from 5.08 to 29.70 μmol TE/g of FW, respectively (Table 2). When expressed according to dry matter, values ranged from 3.55 (sample 2) to 4.63 (sample 3) GAE/g and from 18.73 (sample 2) to 32.15 (sample 4) μmol TE/g for TPC and ABTS, respectively. A significant decrease in sample 2 vs. sample 1 was observed in both ABTS and TPC (Figure 4A,B), which may be due to the loss of natural antioxidants from cane juice. Subsequently, a significant increase was observed in sample 3 (*p* < 0.05) in relation to samples 1 and 2, whilst no significant differences (*p* > 0.05) were observed between sample 3 and the panela. This suggests that antioxidant compounds derived from the heat treatment are generated mainly during the concentration stage and remain constant in the final product (block panela). There are no reports in the literature regarding the evolution of antioxidant activity measured as the TPC and ABTS during panela production. The values found in the present study are within the range reported by Gómez-Narváez et al. [24] for block and granulated panela samples (1.1–6.2 mg of GAE/g and 12.7–50.5 μmol TE/g). However, lower values of the TPC (0.26 mg GAE/g) have been reported by Payet et al. [34], whereas much higher values (165–321 mg GAE/g) have been described by Lee et al. [27]. These differences may be due to different varieties of cane being used, alongside different panela processes, antioxidant extraction methods and measurement protocols. According to Payet et al. [34], antioxidant phenolics and flavonoids from the sugarcane stalk are retained in brown sugar during the non-centrifugation procedure. On the other hand, the high temperatures applied during the evaporation process promote non-enzymatic browning reactions, the formation of dark-colored substances with antioxidant activity and greater accessibility of phenolic compounds trapped in complex structures [35]. In this respect, significant correlations were observed between the ABTS results in relation to acrylamide (r^2^ = 0.8517, *p* = 0.0073) and browning (r^2^ = 0.7230, *p* = 0.0427). Figure 4 shows the evolution of antioxidant capacity measured as ABTS (4A) and the TPC (4B) during panela production (results expressed according to dry matter). A similar pattern is observed in both parameters, with a large increase being evident between sample 2 and sample 3, which is confirmed by a significant correlation between these samples (r^2^ = 0.9341, *p* = 0.0007). This fact suggests the essential contribution of polyphenolic compounds to increase ABTS values.

## 4. Conclusions

The formation of acrylamide, HMF and furfural, the evolution of the antioxidant capacity and browning during different stages of panela production were evaluated by applying robust and validated analytical methods for the first time in the present study. Acrylamide occurs mainly at the stage of honey concentration and at the final stage of block panela manufacturing. Antioxidant compounds contributing to antioxidant capacity are mainly generated at the stage of honey concentration, whilst HMF, furfural and soluble compounds contributing to browning are primarily formed during the final stage of the process. The high temperatures applied during the evaporation process promote non-enzymatic browning reactions and the formation of dark-colored substances with antioxidant activity. Acrylamide formation during the final stage (block panela) takes place without interference from reducing sugars, possibly due to the presence of Maillard reaction intermediate compounds, which have been formed in previous stages of the process. These results provide greater insight into the identification of critical points in the process and suggest that mitigation strategies could be focused on the last stage of panela production, which would help panela producers and food safety bodies to control the formation of processing contaminants in this foodstuff.

## Figures and Tables

**Figure 1 foods-09-00531-f001:**
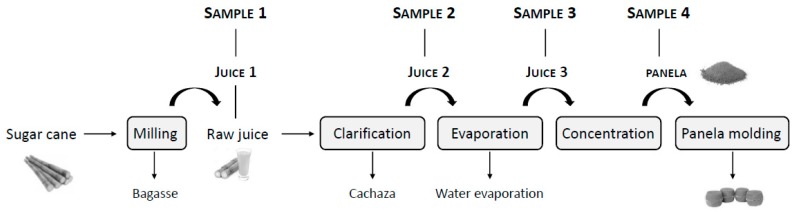
Scheme of the panela production process.

**Figure 2 foods-09-00531-f002:**
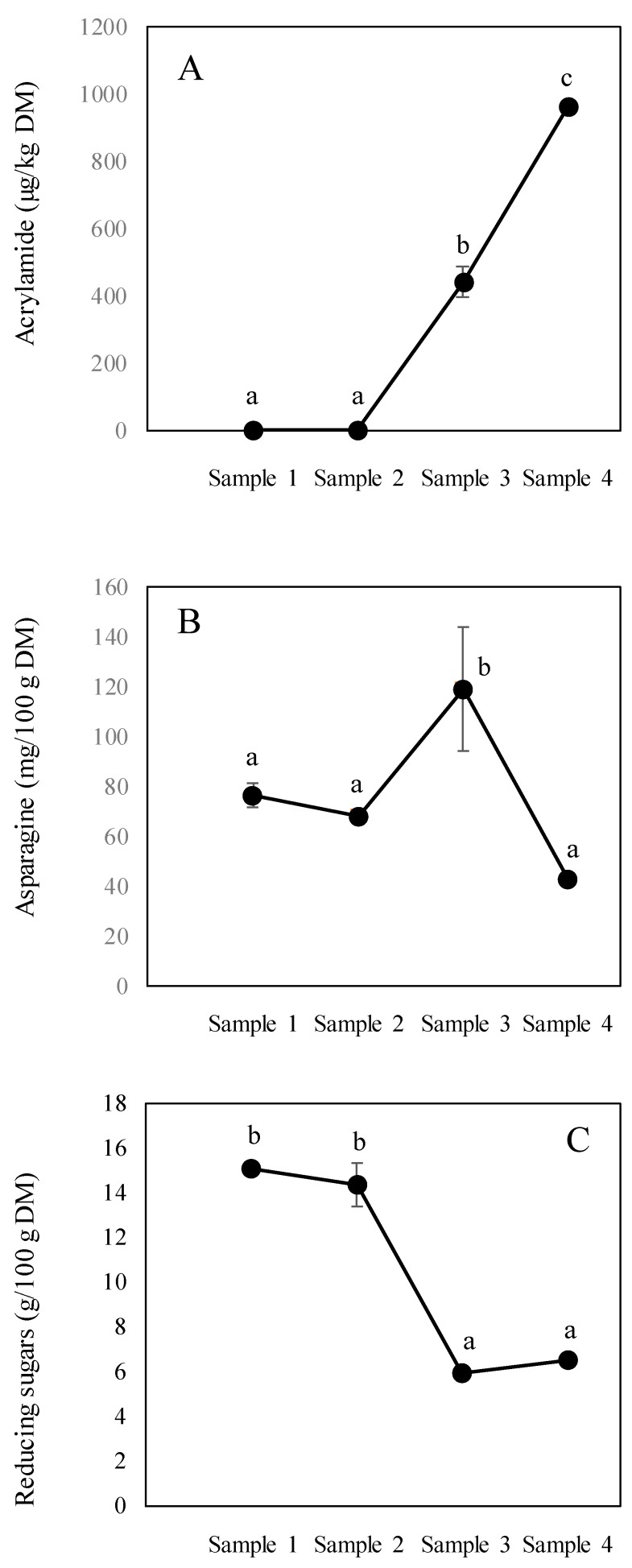
Development of acrylamide (**A**), asparagine (**B**) and reducing sugars (**C**) at different stages of panela production. All data expressed in dry matter (DM). Different letters indicate significant differences (*p* < 0.05).

**Figure 3 foods-09-00531-f003:**
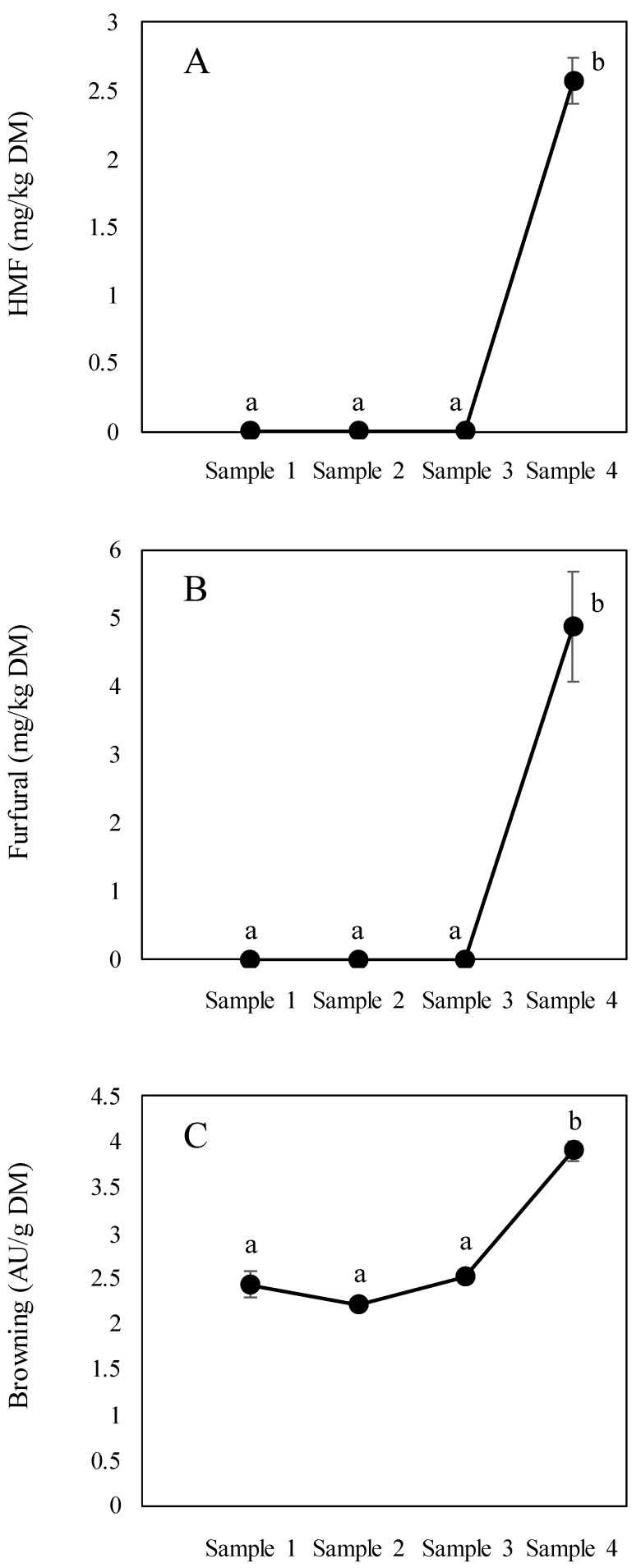
Development of HMF (**A**), furfural (**B**) and browning (420 nm) (**C**) during different stages of panela production. All data expressed in DM. Different letters indicate significant differences (*p* < 0.05).

**Figure 4 foods-09-00531-f004:**
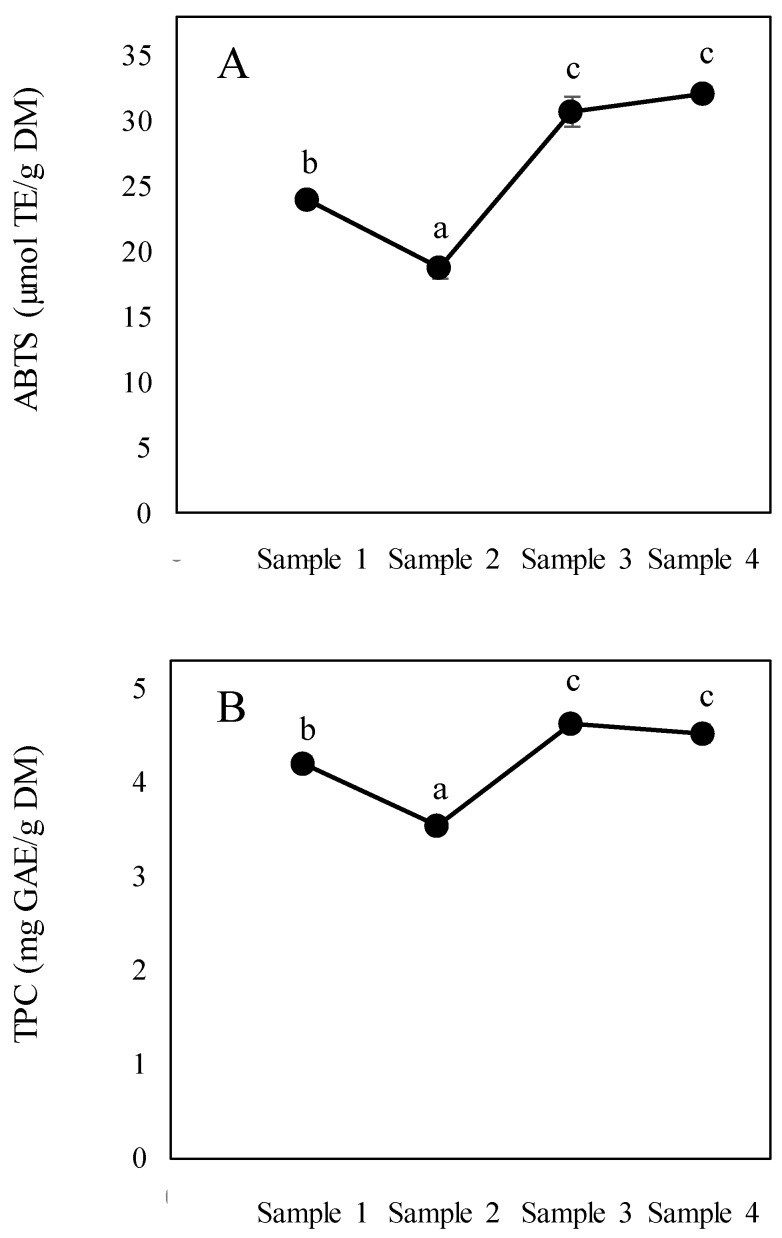
Evolution of antioxidant activity measured as ABTS (2,2’-azino-bis(3-ethylbenzothiazoline-6-sulfonic acid (**A**) and TPC (Total Phenolic Content) (**B**) during different stages of panela production. All data expressed in DM. Different letters indicate significant differences (*p* < 0.05).

**Table 1 foods-09-00531-t001:** Moisture, soluble solids, pH and acrylamide precursors (reducing sugars and asparagine) in samples obtained during panela processing (fresh weight—FW).

	Sample 1	Sample 2	Sample 3	Sample 4
**Moisture (%)**	78.8 ^c^	81.5	32.5	7.60
**Soluble solids (° Brix)**	21.2	18.5	67.5	92.4
**pH**	5.40	5.50	5.70	6.00
**Reducing sugars (g/100 g)**	3.20 ^b^ (3.18–3.22)	2.65 ^a^ (2.53–2.78)	4.01 ^c^ (3.89–4.12)	6.02 ^d^ (6.02–6.03)
**Asparagine (mg/100 g)**	16.2 ^a^ (15.5–17.0)	12.7 ^a^ (12.4–12.9)	80.5 ^c^ (68.7–92.3)	39.7 ^b^ (38.4–41.0)

Results are expressed as mean (range) (*n* = 2). Different superscripts in the same row indicate significant differences (*p* < 0.05).

**Table 2 foods-09-00531-t002:** Acrylamide, hydroxymethylfurfural (HMF) and furfural content, browning (420 nm), ABTS (2,2’-azino-bis(3-ethylbenzothiazoline-6-sulfonic acid) and total phenolic content (TPC) in samples obtained during panela processing (FW).

	Sample 1	Sample 2	Sample 3	Sample 4
**Acrylamide (µg/kg)**	<LOQ	<LOQ	298 ± 31 ^a^	890 ± 15 ^b^
**HMF (mg/kg)**	<LOQ	<LOQ	<LOQ	2.37 ± 0.16
**Furfural (mg/kg)**	<LOQ	<LOQ	<LOQ	4.50 ± 0.75
**Browning (AU/g)**	0.51 ^a^ (0.50–0.53)	0.41 ^a^ (0.40–0.41)	1.70 ^b^ (1.69–1.70)	3.60 ^c^ (3.53–3.67)
**ABTS (µmol TE/g)**	5.08 ± 0.13 ^b^	3.47 ± 0.15 ^a^	20.68 ± 0.78 ^c^	29.70 ± 0.41 ^d^
**TPC (mg GAE/g)**	0.89 ± 0.02 ^b^	0.66 ± 0.01 ^a^	3.12 ± 0.04 ^c^	4.18 ± 0.08 ^d^

Results are expressed as mean (range) (*n* = 2) and as mean ± SD (*n* = 3). Different superscripts in the same row indicate significant differences (*p* < 0.05).

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
