# Peer review of "Formation of Acrylamide and other Heat-Induced Compounds during Panela Production"

_foods, 2020, doi:10.3390/foods9040531_

Round 1

Reviewer 1 Report

Presentes article is partially interesting. However I have some comments and remarks

Instead of Fig. 1. Authors should put the process of panela production on their own block diagram. Besides, there are a few undefined not English words on Fig. 1, such baggase and bagacera

Lines 185-202 It is not adequate to give a full description of panela production in Results and Discussion section.

Authors should include the statistical evaluation for all data in Table 1.

 Table 2. How did authors determin the significant differences between no numbers data LOQ?

Fig. 2B How is it possible that sample 4 is not statistically different form sample 1? The standards deviation is very small.

Fig. 4. The quality of Fig. 4 is poor.

Author Response

Reviewer 1:

Presentes article is partially interesting. However I have some comments and remarks.

Thank you very much for your comments.

Instead of Fig. 1. Authors should put the process of panela production on their own block diagram. Besides, there are a few undefined not English words on Fig. 1, such baggase and bagacera.

According to the referee’s comment, Figure 1 has been removed and replaced by our own block diagram. The term “baggase” has been replaced by “bagasse”.

Lines 185-202. It is not adequate to give a full description of panela production in Results and Discussion section.

We think that some description of the process is necessary; however, following the suggestion of the reviewer, this paragraph has been rewritten and shortened.

Authors should include the statistical evaluation for all data in Table 1.

Authors agree with referee. Statistical evaluation for moisture data was included. In pH and soluble solids, only one measurement was made until the reading was constant. This was clarified in material and methods.

Table 2. How did authors determine the significant differences between no numbers data LOQ?

Authors agree with referee. Statistical results have been checked and included in revised Table 2.

Fig. 2B How is it possible that sample 4 is not statistically different form sample 1? The standards deviation is very small.

For the evaluation of the statistical differences between groups, LSD (Least Significant Difference) test was applied. According to the results obtained and despite the small standard deviations, differences between sample 1 and 4 were not significant (p > 0.05). The application of Tukey-HSD, Scheffe, Bonferroni and Duncan tests indicated the same results.

Fig. 4. The quality of Fig. 4 is poor.

Quality of Fig. 4 has been improved.

Reviewer 2 Report

I have carefully read the manuscript and found that this work is important for the development of the pure and applied research. In my opinion, manuscript entitled: “Formation of acrylamide and other heat-induced compounds during panela production” might be of interest to a specific group of readers. The authors presented a very interesting and comprehensive problem of occurrence of acrylamide and other heat-induced compounds i. e. HMF or furfural during panela production. The work is innovative, the authors describe the methods used during the research very well, and the work itself complements previously published research entitled: “Occurrence of acrylamide and other heat-induced compounds in panela: Relationship with physicochemical and antioxidant parameters” - Food Chemistry, 2019, 301, 125256. However, there are some improvements (only minor) that should be corrected.

# The introduction should better describe the industrial use of panela.

# The introduction should include a description of HMF toxicity and furfural (mechanism of toxicity to cells) to better present the research problem and document the validity of the study.

# Authors should better present the novelty of the presented research and its validity. If possible, the discussion should be developed.

Author Response

Reviewer 2:

I have carefully read the manuscript and found that this work is important for the development of the pure and applied research. In my opinion, manuscript entitled: “Formation of acrylamide and other heat-induced compounds during panela production” might be of interest to a specific group of readers. The authors presented a very interesting and comprehensive problem of occurrence of acrylamide and other heat-induced compounds i. e. HMF or furfural during panela production. The work is innovative, the authors describe the methods used during the research very well, and the work itself complements previously published research entitled: “Occurrence of acrylamide and other heat-induced compounds in panela: Relationship with physicochemical and antioxidant parameters” - Food Chemistry, 2019, 301, 125256. However, there are some improvements (only minor) that should be corrected.

Thank you very much for your comments.

The introduction should better describe the industrial use of panela.

The industrial use of panela was included.

The introduction should include a description of HMF toxicity and furfural (mechanism of toxicity to cells) to better present the research problem and document the validity of the study.

We appreciate this comment. Toxicity about HMF and furfural has been described in more detail.

Authors should better present the novelty of the presented research and its validity. If possible, the discussion should be developed.

Thanks for this comments. These studies on heat-induced process contaminants are pioneers in panela processing. In consequence, there are not many previous studies applying validated methodologies in the literature to compare the results. Referee’s comment has been taken into account and the novelty of this research has been mentioned in the objective and the conclusions of the research.

Reviewer 3 Report

The aim of this study is to investigate the formation of heat induced compounds (acrylamide, hydroxymethyl furfural and furfural) during panela production by validated analytical methods. The authors in addition to the investigation of acrylamide, furfural and hydroxymethylfurfural formation, also evaluated the browning, the changes in antioxidant capacity  and total phenolic contents at different stages of panela production. The results point out the importance of heating conditions applied during panela production and reveals the critical points needs to be controlled, therefore provides an advance in the current knowledge. However there are some minor points need to be taken into consideration:

  • Some of the results were given per fresh weight of the samples. The results should be reported for dry matter of the samples; since the samples differ to great extent in terms of moisture content.
  • More explanation is needed to prove that panela is not only important for Colombia. Otherwise, the significance of the study to the broad range of readers is questionable.

Here I would like to give some detailed comments:

Lines 12-14: “TPC” should be written outside of brackets. Rewrite the sentence, it seems antioxidant capacity and TPC are in the same group of heat-induced compounds...

Lines 29-31: Reconsider the production scheme: grinding of sugar cane should be the first step.

Paragraph 33-40: Is it only particular importance to Colombia? If not, please explain the consumption of panela in other countries in the world, emphasizing the importance of this product to the world.

Line 48: Add “amino acid” term after free; it should be free amino acids…

Lines 49-51: Rewrite the sentence.

Lines 61-62: Change “by applying a robust…..” with “by applying robust and validated analytical methods”.

Lines 88-89: Is it possible to determine moisture content of samples with high sugar content at this high temperature? You should have used a vacuum oven for this purpose. Please explain your reasoning.

Table 1: Are the values for moisture, soluble solids and pH average values? Add standard devaiations for these values.

Lines 209-211: Please do not use the term "formation rate", since this is not kinetic data.

Lines 211-212: In principle, the results should be given in dry matter. Otherwise the increase in acrylamide concentration is partly due to the decrease of moisture content. Even if it will not change the trend in acrylamide concentration between samples, it is not correct to compare samples while there is big difference in moisture content.

Line 230: 140 µg/kg raw sugar or raw sugar cane?

Lines 250-252: Apart from this sentence, there is no information about how Maillard Reaction progresses and how acrylamide is formed. It will be helpful if this information is added in the introduction section.

Lines 263-264: Values should be given in dry matter.

Lines 274-275: Glucose and fructose are hexoses. What does this sentence mean?

Line 330: Change “bock panela” with “block panela”.

Author Response

Reviewer 3:

The aim of this study is to investigate the formation of heat induced compounds (acrylamide, hydroxymethyl furfural and furfural) during panela production by validated analytical methods. The authors in addition to the investigation of acrylamide, furfural and hydroxymethylfurfural formation, also evaluated the browning, the changes in antioxidant capacity and total phenolic contents at different stages of panela production. The results point out the importance of heating conditions applied during panela production and reveals the critical points needs to be controlled, therefore provides an advance in the current knowledge. However there are some minor points need to be taken into consideration:

Thank you very much for your comments.

Some of the results were given per fresh weight of the samples. The results should be reported for dry matter of the samples; since the samples differ to great extent in terms of moisture content.

Authors agree with reviewer. Samples differ to great extent in terms of moisture content and because of that, final results (included in Figures) are expressed as dry matter in order to compare the net formation in each stage. However, authors consider interesting also including some results expressed as fresh matter with the aim of providing more information to researchers, producers and the official control about the progress of the samples.

More explanation is needed to prove that panela is not only important for Colombia. Otherwise, the significance of the study to the broad range of readers is questionable.

More information about panela production over the world has been included in the revised manuscript. Lines 48-55 described which are the principal countries for the world production of panela as well as for its industrial use.

Here I would like to give some detailed comments:

Lines 12-14: “TPC” should be written outside of brackets. Rewrite the sentence, it seems antioxidant capacity and TPC are in the same group of heat-induced compounds...

Sentence has been rewritten.

Lines 29-31: Reconsider the production scheme: grinding of sugar cane should be the first step.

Sentence has been rewritten and process is described according to the scheme from Figure 1.

Paragraph 33-40: Is it only particular importance to Colombia? If not, please explain the consumption of panela in other countries in the world, emphasizing the importance of this product to the world.

As mentioned before, lines 48-55 include information about panela production over the world.

Line 48: Add “amino acid” term after free; it should be free amino acids…

Authors apologize for the mistake. Amino acid has been included in the sentence.

Lines 49-51: Rewrite the sentence.

Sentence has been rewritten.

Lines 61-62: Change “by applying a robust…..” with “by applying robust and validated analytical methods”.

Change has been done.

Lines 88-89: Is it possible to determine moisture content of samples with high sugar content at this high temperature? You should have used a vacuum oven for this purpose. Please explain your reasoning.

The analysis was carried out following specific panela methodologies. The methodology was clarified and the specific cite was included.

Table 1: Are the values for moisture, soluble solids and pH average values? Add standard deviations for these values.

Moistures standard deviations was added. In pH and soluble solids, only one measurement was made until the reading was constant. This was clarified in material and methods.

Lines 209-211: Please do not use the term "formation rate", since this is not kinetic data.

Change has been done.

Lines 211-212: In principle, the results should be given in dry matter. Otherwise the increase in acrylamide concentration is partly due to the decrease of moisture content. Even if it will not change the trend in acrylamide concentration between samples, it is not correct to compare samples while there is big difference in moisture content.

Authors agree with reviewer. As mentioned before, samples differ to great extent in terms of moisture content and because of that, final results (included in Figures) are expressed as dry matter in order to compare the samples. However, authors consider interesting also including some results expressed as fresh matter with the aim of providing more information about the samples and what happen during the panela production. We think this is important since very few information can be found in the literature concerning the production process and on this way we make these data available for others.

Line 230: 140 µg/kg raw sugar or raw sugar cane?

Authors analyzed raw sugar.

Lines 250-252: Apart from this sentence, there is no information about how Maillard Reaction progresses and how acrylamide is formed. It will be helpful if this information is added in the introduction section.

Introduction has been modified and more information about Maillard reaction has been included.

Lines 263-264: Values should be given in dry matter.

Results expressed as dry matter has been included.

Lines 274-275: Glucose and fructose are hexoses. What does this sentence mean?

Sentence describes that furfural is generated from pentoses and not generated from hexoses (glucose and fructose). Sentence has been modified in order to better understand the meaning.

Line 330: Change “bock panela” with “block panela”.

Block panela has been corrected.

Round 2

Reviewer 1 Report

Article can be accepted in this form

Author Response

Answer to the comments point by point

Reviewer 3:

-please add the line numbers of your revisions (revised manuscript R1).

We want to apologize because we have noticed that in the edited version of our manuscript (R1) made by the journal in its typical template the red font that we used to mark corrections was not respected. Now we have included again the red font for the corrections and the reviewer 3 can check that we followed his/her instructions during R1.

-lines 48-55: no highlighted/revised text?

As previously mentioned, now we have marked again in red the corrected paragraph so that the reviewer can check that revision suggested was performed.

-lines 250-252: no highlighted/revised text in the Introduction addressing this comment?

This is again the same problem due to the elimination of the red font while editing the text. In the current version of the manuscript, the reviewer can check the Introduction was modified and more information about Maillard reaction was included. We apologize.

-2.9.1: TPC is in principle another antioxidant capacity assay (measuring the reducing capacity)?

TPC, Total Phenolic Content, measured according to the Folin-Ciocalteu method, is not an antioxidant capacity assay per se but a method to analyse the total presence of phenolic compounds, which are considered one of the main responsible for the antioxidant activity of vegetable foods. We measured them trying to find responsible for the antioxidant activity detected with the ABTS assay.

Trying to clarify that TPC is not another antioxidant capacity assay, we have re-written some short sentences in the abtract and in the discussion (Lines 301-304) to help to differentiate.

-ABTS assay: data usually as TE (Trolox equivalents).

We agree with the reviewer that Trolox equivalents is the more frequent form to express the results of the ABTS assay. However, gallic acid equivalents are also used in the case of vegetable foods according to their composition, this is the case of coffee (see Opitz et al., 2014, Chapter 26 – Methodology for the measurement of antioxidant capacity of coffee: a validated platform composed of three complementary antioxidant assays, In Processing and Impact on Antioxidants in beverages, pag. 253-264). For panela, based on its composition in phenolic acids, gallic acid is considered as a good standard to express the antioxidant ability (i.e. Payet et al., 2005, J. Agric Food Chem., 53, 10074-10079).

-Table 1 and Table 2: 3 significant digits are usually sufficient; SD: usually n=3 as the minimum to calculate the SD; range if only n=2.

Thanks for this suggestions. According with these new comments, Tables 1 and 2 have been modified to be statistically consistent.

-lines 274-275, revised manuscript l.280:  "Since furfural is generated from pentoses and non-hexoses..." - does this mean that furfural can also be generated from non-hexoses other than pentoses? Please see your response to lines 274-275 and rephrase as appropriate.

We apologise for this mistake. In the present version of the manuscript we have amended as follows: Lines 280-281 “Since furfural is generated from pentoses and not from hexoses, it may be hypothesized that furfural is formed from the interconversion of HMF”

-a final English check of the revised manuscript by a native English speaker is also recommended.

This manuscript was yet submitted to English edition by a native English speaker. During this second revision, an extensive English correction has been performed and mistakes detected have been corrected.

We thank editor and reviewers the opportunity for submmiting a new revised version of our manuscript that we hope will reach the quality standards for publication in Foods.
